# Category Anchor-Guided Unsupervised Domain Adaptation for Semantic Segmentation

**Qiming Zhang**[*1]   **Jing Zhang**[*1]   **Wei Liu**[2]   **Dacheng Tao**[1]

[1]UBTECH Sydney AI Centre, School of Computer Science, Faculty of Engineering
The University of Sydney, Darlington, NSW 2008, Australia
[2]Tencent AI Lab, China
qzha2506@uni.sydney.edu.au, jing.zhang1@sydney.edu.au
wl2223@columbia.edu, dacheng.tao@sydney.edu.au

## Abstract

Unsupervised domain adaptation (UDA) aims to enhance the generalization capability of a certain model from a source domain to a target domain. UDA is of particular significance since no extra effort is devoted to annotating target domain samples. However, the different data distributions in the two domains, or *domain shift/discrepancy*, inevitably compromise the UDA performance. Although there has been a progress in matching the marginal distributions between two domains, the classifier favors the source domain features and makes incorrect predictions on the target domain due to category-agnostic feature alignment. In this paper, we propose a novel category anchor-guided (CAG) UDA model for semantic segmentation, which explicitly enforces category-aware feature alignment to learn shared discriminative features and classifiers simultaneously. First, the category-wise centroids of the source domain features are used as guided anchors to identify the active features in the target domain and also assign them pseudo-labels. Then, we leverage an anchor-based pixel-level distance loss and a discriminative loss to drive the intra-category features closer and the inter-category features further apart, respectively. Finally, we devise a stagewise training mechanism to reduce the error accumulation and adapt the proposed model progressively. Experiments on both the GTA5→Cityscapes and SYNTHIA→Cityscapes scenarios demonstrate the superiority of our CAG-UDA model over the state-of-the-art methods. The code is available at `https://github.com/RogerZhangzz/CAG_UDA`.

## 1   Introduction

Semantic segmentation is a classical computer vision task that refers to assigning pixel-wise category labels to a given image to facilitate downstream applications such as autonomous driving, video surveillance, and image editing. The recent progress in semantic segmentation has been dominated by deep neural networks trained on large datasets. Despite their success, annotating labels at the pixel level is prohibitively expensive and time-consuming, *e.g.*, about 90 minutes for a single image in the Cityscapes dataset [8]. One economical alternative is to exploit computer graphics techniques to simulate a virtual 3D environment and automatically generate images and labels, *e.g.*, GTA5 [31] and SYNTHIA [32]. Although synthetic images have similar appearances to real images, there still exist subtle differences in textures, layouts, colors, and illumination conditions [11, 42–44], which result in different data distributions, or *domain discrepancy*. Consequently, the performance of a certain model trained on synthetic datasets degrades drastically when applied to realistic scenes. To address this issue, one promising approach is domain adaptation [1, 45, 15, 34, 36, 27, 33, 40, 47, 13] to

---

[*]indicates equal contributions.

reduce the domain shift and learn a shared discriminative model for both domains. In this paper, we tackle the more challenging unsupervised domain adaptation (UDA) situation, where no labels are available in the target domain during training.

Previous methods have tried to learn domain-invariant representations by matching the distributions between source and target domains at the appearance level [27, 34, 40, 13, 21], feature level [14, 27, 3, 13], or output level [45, 36, 26]. However, even though matching the global marginal distributions can bring the two domains closer, $e.g.$, reaching a lower maximum mean discrepancy (MMD) [25] or a saddle point in the minimax game via adversarial learning [13], it does not guarantee that samples from different categories in the target domain are properly separated, hence compromising the generalization ability. To tackle this issue, one could instead consider category-aware feature alignment by matching the local joint distributions of features and categories [7, 19, 33]. Other approaches adopt the idea of self-training by generating pseudo-labels for samples in the target domain and providing extra supervision to the classifier [47, 21, 3]. Together with supervision from the source domain, this enforces the network to simultaneously learn domain-invariant discriminative feature representations and shared decision boundaries through back-propagation. The ideas of minimizing the entropy (uncertainty) of the output [39] or discrepancies between the outputs of two classifiers (voters) [26] have also been exploited to implicitly enforce category-level alignment.

Although category-level alignment and self-training methods have produced some promising results, there are still some outstanding issues that need to be addressed to further improve the adaptation performance. For example, error-prone pseudo-labels will mislead the classifier and accumulate errors. Meanwhile, implicit category-level alignment may be affected by category imbalance. To deal with these issues and take advantage of both approaches, here we propose a novel idea of *category anchors*, which facilitate both category-wise feature alignment and self-training. It is motivated by the observation that features from the same category tend to be clustered together. Moreover, the centroids of source domain features in each category can serve as explicit anchors to guide adaptation.

Specifically, we propose a novel category anchor-guided unsupervised domain adaptation model (CAG-UDA) for semantic segmentation. This model explicitly enforces category-wise feature alignment to learn shared feature representations and classifiers for both domains simultaneously. First, the centroids of category-wise features in the source domain are used as anchors to identify the active features in the target domain. Then, we assign pseudo-labels to these active features according to the category of the closest anchor. Lastly, two loss functions are proposed: the first is a pixel-level distance loss between the guiding anchors and active features, which pushes them closer and explicitly minimizes the intra-category feature variance; the other is a pixel-level discriminative loss to supervise the classifier and maximize the inter-category feature variance. To reduce the error accumulation of incorrect pseudo-labels, we propose a stagewise training mechanism to adapt the model progressively.

The main contributions of this paper can be summarized as follows. First, we propose a novel category anchor idea to tackle the challenging UDA problem in semantic segmentation. Second, we propose a simple yet effective category anchor-based method to identify active features in the target domain, further enabling category-wise feature alignment. Finally, the proposed CAG-UDA model achieves new state-of-the-art performance in both GTA5→Cityscapes and SYNTHIA→Cityscapes scenarios.

## 2   Related Work

Many recent advances in computer vision [20, 12, 30, 11, 24, 46, 5] have been based on deep neural networks trained on large-scale labeled datasets such as ImageNet [9], Pascal VOC [10], MS COCO [22], and Cityscapes [8]. However, a domain shift between training data and testing data impairs model performance [29, 17, 18]. To overcome this issue, a variety of domain adaptation methods for classification [6, 23, 37, 28, 41, 3, 19], detection [38, 16], and segmentation [7, 14, 13, 27, 34, 40, 21, 47] have been proposed. In this paper, we focus on the challenging semantic segmentation problem. The current mainstream approaches include style transfer [27, 34, 40, 13, 21], feature alignment [7, 14, 13], and self-training [47, 21]. As our work is most related to the latter two approaches, we briefly review and discuss their characteristics.

*Feature distribution alignment*: Previous methods that match the global marginal distributions between two domains [14, 13, 27] do not distinguish local category-wise feature distribution shifts. Consequently, error-prone predictions are made for misaligned features with shared decision bound-

aries. In contrast to these methods, we propose a category-wise feature alignment method to explicitly reduce category-level mismatches and learn discriminative domain-invariant features. The idea of category-level feature alignment was also exploited in [26, 33] for semantic segmentation. Luo *et al.* proposed a weighted adversarial learning method to align the category-level feature distributions implicitly [26]. Saito *et al.* tried to align the feature distributions and learn discriminative domain-invariant features by utilizing task-specific classifiers as a discriminator [33]. In contrast to the implicit feature alignment in the aforementioned methods, we propose a novel category anchor-guided method, which directly aligns category-wise features in both domains.

*Pseudo-label assignment*: Assigning pseudo-labels to target domain samples based on the trained classifiers helps adapt the feature extractor and classifier to the target domain. Zou *et al.* [47] proposed an iterative self-training UDA model by alternatively generating pseudo-labels and retraining the model. They also dealt with the category imbalance issue by controlling the proportion of selected pseudo-labels in each category [47]. Li *et al.* [21] proposed a bidirectional learning domain adaptation model that alternately trains the image translation model and the self-supervised segmentation adaptation model. In contrast to these methods, where pseudo-labels were determined according to the predicted category probability, we propose a category anchor-based method to generate trustable pseudo-labels. Compared with selected samples that have been "correctly" classified with high confidence, our selected samples are not determined by the decision boundaries so are more *informative* for the classifier to further adapt to the target domain.

The idea of assigning pseudo-labels based on category centers has also been utilized in domain adaptation for classification, *e.g.*, category centroids in [41], prototypes in [3], and cluster centers in [19]. The former two methods minimize the distance loss against category centroids, while the third minimizes contrastive domain discrepancies. Our method differs from these methods in several ways. First, we tackle the more challenging task of image semantic segmentation rather than image classification, where dense pixel-wise labels need to be predicted as not just single labels for entire images. Second, we fix the category centroids (hence called *category anchors*) instead of updating them at each iteration. On one hand, the mini-batch size used for segmentation (*e.g.*, 1) in this paper is much smaller than that used for classification. On the other hand, pixels are spatially coherent in an image, so the category centroids calculated at each iteration will be biased and unreliable due to the dominance of homogeneous features. Third, the pseudo-labels of target domain samples are determined by their distance against the category centroids from the source domain instead of the target domain. This is reasonable since: 1) the source domain category centroids are calculated from all training samples based on ground-truth labels, which are reliable; 2) driving the target domain features towards the source domain category centroids can effectively reduce the domain discrepancy. Fourth, together with the category anchor-based distance loss, we also add the segmentation loss based on the pseudo-labeled target samples to learn discriminative feature representations and adapt the decision boundaries simultaneously.

## 3 A category anchor-guided UDA model for semantic segmentation

### 3.1 Problem Formulation

**Supervised semantic segmentation:** A semantic segmentation model $M$ can be formulated as a mapping function from the image domain $X$ to the output label domain $Y$:

$$M : X \to Y, \tag{1}$$

which predicts a pixel-wise category label $\hat{y}$ close to the ground-truth annotation $y \in Y$ for a given image $x \in X$. Usually, the segmentation model $M$ is trained in a supervised manner by minimizing the difference between the prediction $\hat{y}$ and its ground-truth $y$ for every training sample $x$. The cross-entropy (CE) loss is widely used as a measurement, which is defined as:

$$L_{CE} = -\sum_{i=1}^{N} \sum_{j=1}^{H \times W} \sum_{c=1}^{C} y_{ijc} log\left(p_{ijc}\right), \tag{2}$$

where $N$ is the number of training images, $H$ and $W$ denote the image size, $j$ is the pixel index, $C$ is the number of categories, $c$ is the category index, $y_{ijc} \in \{0, 1\}$ is the one-hot vector representation of the ground-truth label, *i.e.*, $\forall i, j, \sum_c y_{ijc} = 1$, and $p_{ijc}$ is the predicted category probability by $M$.

**UDA for semantic segmentation:** Generally, a segmentation model trained on a source domain $X_s$ has a limited generalization capability to a target domain $X_t$, when the distributions between $X_s$ and

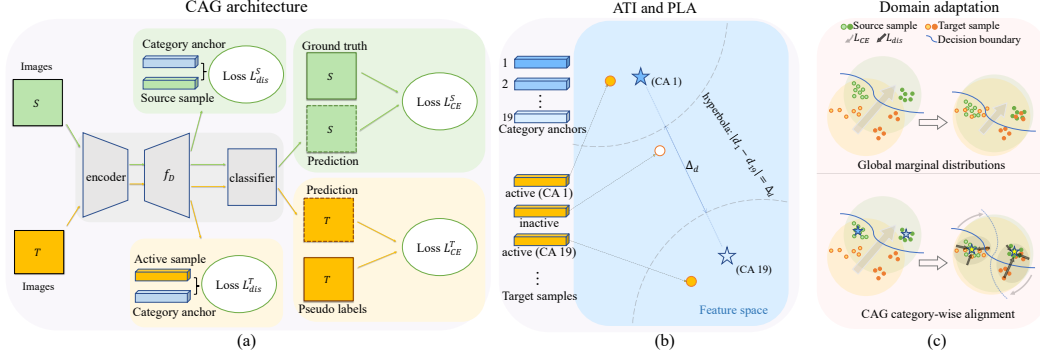

Figure 1: An illustration of the proposed category anchor-guided UDA model for semantic segmentation. (a) The architecture of the proposed CAG-UDA model consists of an encoder, a feature transformer ($f_D$), and a classifier. The green part denotes the source domain flow while the orange parts represent the target domain flow. (b) The illustration of the process of active target sample identification and pseudo label assignment described in Section 3.2. (c) The illustration of the proposed category-wise feature alignment with the anchor-based pixel-level distance loss $L_{dis}$ and cross-entropy loss $L_{CE}$ described in Section 3.3. Best viewed in color.

$X_t$ are different, *i.e.*, there is a domain shift/discrepancy. Several unsupervised domain adaptation models have been proposed, which can be formulated as the following mapping function:

$$M_{uda} : X_s \cup X_t \to Y_s \cup Y_t, \tag{3}$$

where $M_{uda}$ is trained on the labeled training samples $(X_s, Y_s)$ in the source domain together with the training unlabeled samples $X_t$ in the target domain. Typically, the aforementioned CE loss and some domain-adaptation losses are used to align the distributions of both domains (*e.g.*, $p(X_s)$ and $p(Y_s)$) and to learn domain-invariant discriminative feature representations.

**Model components:** The main semantic segmentation approaches have been based on fully convolutional neural networks (CNNs) since the seminal work in [24]. Usually, a DCNN-based model has two parts: an encoder $Enc$ and a decoder $Dec$, where the encoder maps the input image into a low-dimensional feature space and then the decoder decodes it to the label space. The decoder can be further divided into a feature transformation net $f_D$ and a classifier $Cls$, where $Cls$ denotes the last classification layer and $f_D$ denotes the remaining part in $Dec$. Typical encoders are the classification networks pretrained on ImageNet [9], *e.g.*, VGGNet [35] and ResNet [12]. The decoder consists of convolutional layers responsible for context modeling, multi-scale feature fusion, *etc*. UDA methods typically employ a segmentation model with carefully designed modules for domain adaptation.

### 3.2 Network Architecture

The network architecture of our proposed CAG-UDA model is shown in Figure 1(a). The CAG-UDA model employs Deeplab v2 [4] as the base segmentation model, where ResNet-101 is used as the encoder $Enc$ and the ASPP module is used in the decoder $Dec$. To reduce the domain shift, we devise a category anchor-guided alignment module on the features from $f_D$, consisting of category anchor construction (CAC), active target sample identification (ATI), and pseudo-label assignment (PLA) as shown in Figure 1(b). The details are as follows.

**Category anchor construction (CAC):** Based on the observation that pixels in the same category cluster in the feature space, we propose to calculate the centroids of the features of each category in the source domain as a representative of the feature distribution, *i.e.*, the mean. Considering that the features fed into the classifier directly relate to the decision boundaries, we choose the features from $f_D$ to calculate these centroids. Mathematically, this can be written as:

$$f_c^s = \frac{1}{|\Lambda_c^s|} \sum_{i=1}^{N} \sum_{j}^{H \times W} y_{ijc}^s \left( f_D \left( Enc \left( x_i^s \right) \right) |_j \right), \tag{4}$$

where $\Lambda_c^s$ is the index set of all pixels on the training images in the source domain $X_s$ belonging to the $c^{th}$ category, *i.e.*, $\Lambda_c^s = \{(i, j) | y_{ijc}^s = 1\}$, $|\Lambda_c^s|$ denotes the number of pixels in $\Lambda_c^s$, *i.e.*,

$|\Lambda_c^s| = \sum_{i=1}^{N} \sum_{j}^{H \times W} y_{ijc}$, and $f_D\left(x_i^s\right)|_j$ is the feature vector at index $j$ on the feature map $f_D\left(x_i^s\right)$. It is noteworthy that we calculate the category centroids at the *beginning* of each training stage and then keep them *fixed* during training (we propose a stagewise training mechanism in Section 3.4.). Therefore, we call these centroids category anchors (CAs) in this paper, *i.e.*, $CA = \{f_c^s, c = 1, ..., C\}$.

**Active target sample identification (ATI):** To align the category-wise feature distributions between two domains, we expect that the category centroids from the target domain get closer to the category anchors during training. However, on one hand, target sample labels are unavailable. On the other hand, the calculated centroids on target samples are very unstable at each iteration since the mini-batch size is very small (*i.e.*, 1) in this paper and image pixels are spatially coherent. To tackle these issues, we propose identifying *active* target samples and assigning them pseudo-labels for the subsequent feature alignment. The term "active target samples" refers to target samples near one category anchor and far from the other anchors, *i.e.*, being *activated* by one specific category anchor. Mathematically, this can be formulated as follows. We first define the distance between a target feature $f_D\left(Enc\left(x_i^t\right)\right)|_j$ and the $c^{th}$ category anchor as

$$d_{ijc}^t = \left\| f_c^s - f_D\left(Enc\left(x_i^t\right)\right)|_j \right\|_2, \tag{5}$$

where $\|\cdot\|_2$ is the $L_2$ norm of a vector. Then, we sort $\{d_{ijc}^t, c = 1, ..., C\}$ in an ascending order and compare the shortest distance $d_{ijc^*}^t$ with the second shortest $d_{ijc'}^t$. If their difference is larger than a predefined threshold $\triangle_d$, we identify this target sample as active one, *i.e.*,

$$a_{ij}^t = \begin{cases} 1, & d_{ijc'}^t - d_{ijc^*}^t > \triangle_d, \\ 0, & otherwise, \end{cases} \tag{6}$$

where $a_{ij}^t$ denotes the active state of the target feature $f_D\left(Enc\left(x_i^t\right)\right)|_j$. Like the category anchors, we calculate the active states at the beginning of each training stage and keep them fixed during subsequent stages. This is explained in Section 3.4, where we introduce a stagewise training mechanism.

**Pseudo-label assignment (PLA):** After we obtain the active state according to Eq. (6), a pseudo label $c^*$ can be assigned to $x_i^t$ according to its closest category anchor $f_{c^*}^s$ with a reliable margin $\triangle_d$:

$$\hat{y}_{ijc^*}^t = 1, if\ d_{ijc^*}^t < d_{ijc}^t - \triangle_d, \forall c \neq c^*. \tag{7}$$

Due to the lack of the target domain labels, the classifier layer is biased to the source domain and does not generalize well to the target domain, as shown in Figure 1(c). Consequently, some of the pseudo-labels from predicted probabilities may be error-prone. However, based on the observation of the intra-category clustering characteristics, the generated pseudo-labels via category anchors are independent of the biased classifier and are thus more *reliable* than those assigned by predicted category probabilities. Further, considering that high-probability samples have been "correctly" classified by the classifier layer with high confidence, these samples provide only weak supervision signals. In contrast, active samples are more *informative* for adapting the classifier to the target domain as the classifier layer may not predict these active samples with high probabilities.

## 3.3 Objective Functions

When training the CAG-UDA model, we leverage a CE loss $L_{CE}^s$ as defined in Eq. (2). We also propose a category-wise distance loss $L_{dis}^s$ on the source domain samples and two domain adaptation losses on the active target samples, *i.e.*, a CE loss $L_{CE}^t$ and a category-wise distance loss $L_{dis}^t$ based on the pseudo-labels, to guide the adaptation process. These are defined as:

$$L_{dis}^s = \sum_{i=1}^{N} \sum_{j=1}^{H \times W} \sum_{c=1}^{C} y_{ijc}^s \left\| f_c^s - f_D\left(Enc\left(x_i^s\right)\right)|_j \right\|^2, \tag{8}$$

$$L_{CE}^t = -\sum_{i=1}^{M} \sum_{j=1}^{H \times W} a_{ij}^t \sum_{c=1}^{C} \hat{y}_{ijc}^t log\left(p_{ijc}^t\right), \tag{9}$$

$$L_{dis}^t = \sum_{i=1}^{M} \sum_{j=1}^{H \times W} a_{ij}^t \sum_{c=1}^{C} \hat{y}_{ijc}^t \left\| f_c^s - f_D\left(Enc\left(x_i^t\right)\right)|_j \right\|^2. \tag{10}$$

Although only the active samples are directly driven towards the category anchors by $L_{dis}^t$, other inactive target samples within each category may also follow the active samples due to being clustered. Therefore, minimizing $L_{dis}^t$ indeed reduces the intra-category variances in the target domain. Meanwhile, $L_{CE}^t$ leverages the pseudo-labels to update the network weights together with the source domain CE loss, prompting the encoder, decoder, and classifier to adapt to the target domain and therefore reducing the intra- and inter-category variances simultaneously. The illustration is show in Figure 1(c). To leverage the complementarity between the proposed category anchor-based PLA and category probability-based PLA in [47], we also identify active target samples based on the predicted category probability and add an extra CE loss $L_{CE}^{tP}$ similar to Eq. (9).

$$L_{CE}^{tP} = -\sum_{i=1}^{M} \sum_{j=1}^{H \times W} a_{ij}^{tP} \sum_{c=1}^{C} \hat{y}_{ijc}^{tP} log \left( p_{ijc}^{t} \right), \qquad (11)$$

where $a_{ij}^{tP}$, $y_{ijc}^{tP}$ refer to the probability-based active state and assigned pseudo-labels respectively. Then the final objective function is as follows:

$$L = L_{CE}^s + \lambda_1 \left( L_{dis}^s + L_{dis}^t \right) + \lambda_2 \left( L_{CE}^t + L_{CE}^{tP} \right), \qquad (12)$$

where $\lambda_1$ and $\lambda_2$ are loss weights.

### 3.4 Stagewise Training Procedure

We tried to train the CAG-UDA model in a single stage and update the pseudo-labels at each iteration. However, it is not stable because there are some error-prone pseudo-labels, which may produce incorrect supervision signals, lead to more erroneous pseudo-labels iteratively and trap the network to a local minimum with poor performance eventually, $e.g.$ less than 30 mIoU. To address this issue, we propose a stagewise training mechanism as summarized in Algorithm 1. First, we pretrain the segmentation model on the source domain. Then, we leverage the global feature alignment method in [14] to warm up the training process and obtain a well-initialized model. Next, we train the CAG-UDA model with the proposed losses for several stages. At the beginning of each stage, we calculate the CAs, identify the active target samples, and assign pseudo-labels to them. By using this stagewise delayed updating mechanism, we avoid updating the pseudo-labels at each iteration and reduce the error accumulation. Hence, $L_{dis}^t$ and $L_{CE}^t$ serve as two regularizations on the network.

---

**Algorithm 1** Stagewise training the CAG-UDA model

---

**Input:** training dataset: $(X_s, Y_s, X_t)$, maximum stages: $K$, maximum iterations: $L$, distance threshold: $\triangle_d$.
**Output:** $M_K$ and $(\hat{Y}_s, \hat{Y}_t)$.
 1: Pretraining: $M_0^p \leftarrow (X_s, Y_s)$ according to [4];
 2: Warm-up: $M_0 \leftarrow (X_s, Y_s)$ and $M_0^p$ according to [14];
 3: **for** $k \leftarrow 1 \ to \ K$ **do**
 4:     CAC: $\{f_c^s\} \leftarrow M_{k-1}$ and $(X_s, Y_s)$ according to Eq. (4);
 5:     ATI: $\{d_{ijc}^t\}, \{a_{ij}^t\} \leftarrow M_{k-1}, (X_s, Y_s, X_t), \{f_c^s\}$ and $\triangle_d$ according to Eq. (5) and Eq. (6);
 6:     PLA: $\{\hat{y}_{ijc^*}^t\} \leftarrow \{d_{ijc}^t\}, \triangle_d$ according to Eq. (7);
 7:     **for** $n \leftarrow 1 \ to \ L$ **do**
 8:         SGD: training $M_{k-1}$ on $(X_s, Y_s, X_t, \{\hat{y}_{ijc^*}^t\}, \{f_c^s\}, \{a_{ij}^t\})$ according to Eq.(12);
 9:     **end for**
10:     $M_k \leftarrow M_{k-1}$
11: **end for**
12: Prediction: $(\hat{Y}_s, \hat{Y}_t) \leftarrow (X_s, X_t)$ and $M_K$.

---

## 4 Experiments

### 4.1 Experimental Settings

**Datasets and evaluation metrics:** Following [21], we evaluate the CAG-UDA model in two common scenarios, GTA5[31]→Cityscapes[8] and SYNTHIA[32]→Cityscapes[8]. GTA5 contains

Table 1: Results of the CAG-UDA model and SOTA methods ( GTA5→Cityscapes).

| | road | sidewalk | building | wall | fence | pole | light | sign | vege. | terrace | sky | person | rider | car | truck | bus | train | motor | bike | mIoU |
|---|---|---|---|---|---|---|---|---|---|---|---|---|---|---|---|---|---|---|---|---|
| Source only | 75.8 | 16.8 | 77.2 | 12.5 | 21.0 | 25.5 | 30.1 | 20.1 | 81.3 | 24.6 | 70.3 | 53.8 | 26.4 | 49.9 | 17.2 | 25.9 | 6.5 | 25.3 | 36.0 | 36.6 |
| AdaptSegNet[36] | 86.5 | 25.9 | 79.8 | 22.1 | 20.0 | 23.6 | 33.1 | 21.8 | 81.8 | 25.9 | 75.9 | 57.3 | 26.2 | 76.3 | 29.8 | 32.1 | 7.2 | 29.5 | 32.5 | 41.4 |
| Source only | 69.9 | 22.3 | 75.6 | 15.8 | 20.1 | 18.8 | 28.2 | 17.1 | 75.6 | 8.00 | 73.5 | 55.0 | 2.9 | 66.9 | 34.4 | 30.8 | 0.00 | 18.4 | 0.00 | 33.3 |
| DCAN[40] | 85.0 | 30.8 | 81.3 | 25.8 | 21.2 | 22.2 | 25.4 | 26.6 | 83.4 | 36.7 | 76.2 | 58.9 | 24.9 | 80.7 | 29.5 | 42.9 | 2.50 | 26.9 | 11.6 | 41.7 |
| Source only | 75.8 | 16.8 | 77.2 | 12.5 | 21.0 | 25.5 | 30.1 | 20.1 | 81.3 | 24.6 | 70.3 | 53.8 | 26.4 | 49.9 | 17.2 | 25.9 | 6.5 | 25.3 | 36.0 | 36.6 |
| CLAN[26] | 87.0 | 27.1 | 79.6 | 27.3 | 23.3 | 28.3 | 35.5 | 24.2 | 83.6 | 27.4 | 74.2 | 58.6 | 28.0 | 76.2 | 33.1 | 36.7 | 6.7 | **31.9** | 31.4 | 43.2 |
| AdvEnt[39] | 89.4 | 33.1 | 81.0 | 26.6 | 26.8 | 27.2 | 33.5 | 24.7 | 83.9 | 36.7 | 78.8 | 58.7 | 30.5 | 84.8 | **38.5** | 44.5 | 1.7 | 31.6 | 32.4 | 45.5 |
| DISE[2] | **91.5** | 47.5 | 82.5 | 31.3 | 25.6 | 33.0 | 33.7 | 25.8 | 82.7 | 28.8 | 82.7 | 62.4 | 30.8 | **85.2** | 27.7 | 34.5 | 6.4 | 25.2 | 24.4 | 45.4 |
| Cycada[13, 21] | 86.7 | 35.6 | 80.1 | 19.8 | 17.5 | 38.0 | **39.9** | 41.5 | 82.7 | 27.9 | 73.6 | **64.9** | 19.0 | 65.0 | 12.0 | 28.6 | 4.5 | 31.1 | 42.0 | 42.7 |
| Source only | 69.0 | 12.7 | 69.5 | 9.9 | 19.5 | 22.8 | 31.7 | 15.3 | 73.9 | 11.3 | 67.2 | 54.7 | 23.9 | 53.4 | 29.7 | 4.6 | 11.6 | 26.1 | 32.5 | 33.6 |
| BLF[21] | 91.0 | 44.7 | **84.2** | **34.6** | 27.6 | 30.2 | 36.0 | 36.0 | 85.0 | **43.6** | **83.0** | 58.6 | 31.6 | 83.3 | 35.3 | **49.7** | 3.3 | 28.8 | 35.6 | 48.5 |
| Source only | 69.8 | 25.4 | 74.7 | 11.3 | 18.3 | 24.2 | 35.6 | 23.3 | 72.0 | 14.4 | 65.3 | 58.7 | 29.0 | 53.1 | 14.3 | 19.2 | 7.9 | 15.1 | 16.3 | 34.1 |
| CAG-UDA | 90.4 | **51.6** | 83.8 | 34.2 | **27.8** | **38.4** | 25.3 | **48.4** | **85.4** | 38.2 | 78.1 | 58.6 | **34.6** | 84.7 | 21.9 | 42.7 | **41.1** | 29.3 | **37.2** | **50.2** |

Table 2: Results of the CAG-UDA model on the testing set ( GTA5→Cityscapes).

| | road | sidewalk | building | wall | fence | pole | light | sign | vege. | terrace | sky | person | rider | car | truck | bus | train | motor | bike | mIoU |
|---|---|---|---|---|---|---|---|---|---|---|---|---|---|---|---|---|---|---|---|---|
| CAG-UDA | 93.2 | 57.0 | 85.6 | 35.7 | 25.1 | 37.5 | 30.8 | 45.3 | 87.1 | 50.1 | 89.4 | 62.7 | 40.8 | 87.8 | 18.0 | 32.4 | 34.5 | 34.4 | 35.4 | 51.7 |

24,966 1914×1052-pixel images and has the same 19 category annotations as Cityscapes. SYN-THIA contains 9,400 1914×1052-pixel images and only has 16 common category annotations. Cityscapes is divided into a training set, a validation set, and a testing set. The training set consists of 2,957 2048×1024-pixel images and the validation set contains 500 images at the same resolution. Following common practice, we report the results on the Cityscapes validation set, specifically, the category-wise intersection over union (IoU). Moreover, we also report the mean IoU (mIoU) of all 19 categories in the GTA5→Cityscapes scenario and the 16 common categories in the SYNTHIA→Cityscapes scenario. Some methods [36, 26, 21] only reported mIoU for 13 common categories in the SYNTHIA→Cityscapes scenario, denoted as mIoU* in this paper.

**Implementation details:** In our experiments, training images were randomly cropped to 1280×640 pixels after being randomly resized by ×1 ∼ ×1.5. Due to GPU memory limitations, the batch size was set to 1 and the weights of all batch normalization layers were frozen. In the warm-up phase, we used a CNN-based domain discriminator comprising 5 convolutional layers of kernel size 3×3, filter numbers [64, 128, 256, 512, 1], and stride 2. The first three convolutional layers are followed by a ReLU layer, while the fourth layer is followed by a leaky ReLU layer parameterized by 0.2. We used a CE loss and an adversarial loss to train the model for 20 epochs. The adversarial loss weights were set to 1e-2. In the stagewise training phase, we trained the CAG-UDA mode for 20 epochs with the SGD optimizer. The initial learning rate was 2.5e-4, which decayed by the poly policy with power 0.9. The weight decay, momentum, $\lambda_1$, and $\lambda_2$ were set to 1e-4, 0.9, 0.3, and 0.7, respectively. $\triangle_d$ was set to 2.5. We also assigned pseudo-labels based on predicted category probabilities, and the threshold $P_0$ was set to 0.95. Experiments were conducted on a TITAN Tesla V100 GPU with PyTorch implementation. Code will be made publicly available.

## 4.2 Main Results

**Quantitative results:** The results of the GTA5→Cityscapes scenario are presented in Table 1 with the best results highlighted in bold. All the models adopted ResNet-101 as a backbone network for fair comparison. Overall, our CAG-UDA model strikingly outperforms all other models with a 50.2 mIoU, surpassing the model trained on the source domain by a significant gain of 16.1. Compared with CLAN [26] and DISE [2], which implicitly align category-level features, our model achieves an extra gain of 4.5 and outperforms them on fence, traffic sign, rider, train, and bike by large margins. This is due to the proposed category anchor-guided alignment method, which explicitly uses category centroids as representatives of feature distributions, reducing the side effect of category imbalance. Like [40, 13], BLF in [21] also involves a style-transfer module but combines it with self-training in a bidirectional learning framework. It achieved the second-best mIoU of 48.5. BLF achieves better results than the CAG-UDA model on stuff categories such as road, building, wall, terrace, and sky but is inferior to the CAG-UDA model for small objects. This is because BLF includes a

Table 3: Results of the CAG-UDA model and SOTA methods ( SYNTHIA→Cityscapes).

| | road | sidewalk | building | wall | fence | pole | light | sign | vegetable | sky | person | rider | car | bus | motor | bike | mIoU | mIoU* |
|---|---|---|---|---|---|---|---|---|---|---|---|---|---|---|---|---|---|---|
| AdaptSegNet[36] | 79.2 | 37.2 | 78.8 | - | - | - | 9.9 | 10.5 | 78.2 | 80.5 | 53.5 | 19.6 | 67.0 | 29.5 | 21.6 | 31.3 | - | 45.9 |
| CLAN[26] | 81.3 | 37.0 | 80.1 | - | - | - | **16.1** | 13.7 | 78.2 | 81.5 | 53.4 | 21.2 | 73.0 | 32.9 | 22.6 | 30.7 | - | 47.8 |
| BLF[21] | 86.0 | 46.7 | 80.3 | - | - | - | 14.1 | 11.6 | 79.2 | 81.3 | 54.1 | 27.9 | 73.7 | **42.2** | **25.7** | 45.3 | - | 51.4 |
| CAG-UDA(13) | 84.8 | 41.7 | **85.5** | - | - | - | 13.7 | **23.0** | 86.5 | 78.1 | **66.3** | 28.1 | 81.8 | 21.8 | 22.9 | **49.0** | - | **52.6** |
| DCAN[40] | 82.8 | 36.4 | 75.7 | 5.1 | 0.1 | 25.8 | 8.0 | 18.7 | 74.7 | 76.9 | 51.1 | 15.9 | 77.7 | 24.8 | 4.1 | 37.3 | 38.4 | - |
| DISE[2] | **91.7** | **53.5** | 77.1 | 2.5 | 0.2 | 27.1 | 6.2 | 7.6 | 78.4 | 81.2 | 55.8 | 19.2 | **82.3** | 30.3 | 17.1 | 34.3 | 41.5 | - |
| AdvEnt[39] | 85.6 | 42.2 | 79.7 | **8.7** | **0.4** | 25.9 | 5.4 | 8.1 | 80.4 | **84.1** | 57.9 | 23.8 | 73.3 | 36.4 | 14.2 | 33.0 | 41.2 | - |
| CAG-UDA(16) | 84.7 | 40.8 | **81.7** | 7.8 | 0.0 | **35.1** | 13.3 | **22.7** | **84.5** | 77.6 | **64.2** | 27.8 | 80.9 | 19.7 | 22.7 | **48.3** | **44.5** | - |

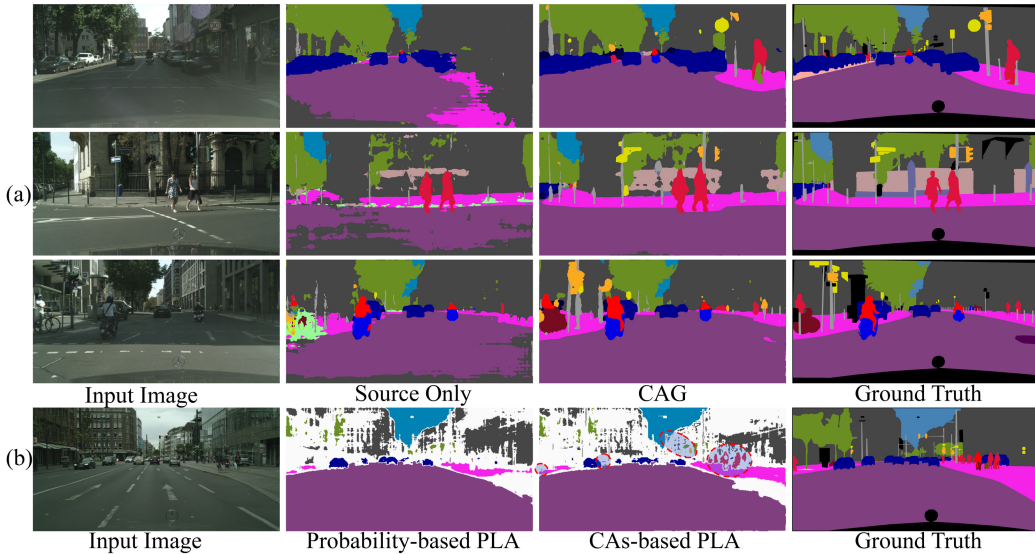

(a) Input Image    Source Only    CAG    Ground Truth

(b) Input Image    Probability-based PLA    CAs-based PLA    Ground Truth

Figure 2: (a) Subjective evaluation of the CAG-UDA model on some images from the Cityscapes validation set. (b) Comparison between probability-based PLA and the proposed CAs-based PLA on an image from the Cityscapes training set. Best viewed in color and zoom-in.

style-transfer module that benefits from the texture clues in the stuff categories and assigns reliable pseudo-labels accordingly. In contrast, CAG-UDA uses a category-anchor guided method that can tackle the category imbalance and generate more informative pseudo-labels, leading to better results on more categories.

We also present the result on the testing set of the Cityscapes dataset in Table 2. The CAG-UDA model reaches 51.7 mIoU, proving the good generalization of our method.

Results in the SYNTHIA→Cityscapes scenario are listed in Table 3. Same as the previous work, we report the performance of the CAG-UDA model in two mIoU metrics: 13 categories (mIoU*) and 16 categories (mIoU) for fair comparisons. Since the domain shift is much larger than the above scenario, the performance is slightly worse. The CAG-UDA model still achieves better results than all previous SOTA methods, including CLAN, BLF, *etc.* Similar to the above discussions with the GTA5 dataset, the superiority of the CAG-UDA model remains in small objects like pole, sign, person, and bike.

**Qualitative results:** Some qualitative segmentation examples are given in Figure 2(a). Training merely on the source domain dataset leads to a limited generalization ability, *e.g.*, the road and person were incorrectly predicted as sidewalk and building in the first row. Benefited from the category anchor-guided adaptation, the proposed CAG-UDA model achieves better results, especially for small objects, *e.g.*, pole, sign, and person. Besides, we also attribute it to the proposed CAs-based pseudo label assignment, which successfully activated small objects and assigned them trustable pseudo-labels, as highlighted in red circles in Figure 2(b). More results can be found in the supplement.

Table 4: Results of ablation study (GTA5→Cityscapes).

| | road | side. | buil. | wall | fenc. | pole | light | sign | vege. | terr. | sky | person | rider | car | truck | bus | train | motor | bike | mIoU | gain |
|---|---|---|---|---|---|---|---|---|---|---|---|---|---|---|---|---|---|---|---|---|---|
| Source only | 69.8 | 25.4 | 74.7 | 11.3 | 18.3 | 24.2 | 35.6 | 23.3 | 72.0 | 14.4 | 65.3 | 58.7 | 29.0 | 53.1 | 14.3 | 19.2 | 7.9 | 15.1 | 16.3 | 34.1 | - |
| Warm-up | 88.4 | 45.2 | 82.0 | 30.1 | 22.0 | 35.4 | 36.7 | 23.7 | 82.7 | 27.6 | 70.8 | 51.4 | 26.9 | 81.5 | 14.5 | 25.0 | 21.4 | 13.0 | 7.9 | 41.4 | 7.3 |
| $+L_{CE}^{tP}$ | 88.8 | 45.5 | 83.7 | 33.2 | 21.4 | 39.5 | **40.0** | 25.9 | 83.9 | 33.8 | 74.3 | 58.2 | 24.9 | 84.8 | 19.3 | 32.8 | 22.6 | 15.0 | 14.7 | 44.3 | 10.2 |
| $+L_{CE}^{t}$ | 88.3 | 46.9 | 81.5 | 28.7 | 27.7 | 38.9 | 27.0 | 40.4 | 83.7 | 31.2 | 74.9 | 61.8 | 30.2 | 84.0 | 15.9 | 36.7 | 23.4 | 23.3 | 31.7 | 46.1 | 12.0 |
| $+L_{dis}^{sP} + L_{dis}^{tP}$ | 89.4 | 40.1 | 81.8 | 31.0 | 22.6 | 39.9 | 41.2 | 23.2 | 83.0 | 28.3 | 68.5 | 54.5 | 23.8 | 85.7 | 21.5 | 25.6 | 0.7 | 13.9 | 8.5 | 41.2 | 7.1 |
| $+L_{dis}^{s} + L_{dis}^{t}$ | 88.9 | 41.7 | 82.0 | 31.7 | 22.5 | 39.7 | 41.2 | 23.5 | 82.7 | 27.0 | 70.0 | 57.8 | 25.7 | **85.8** | **21.9** | 27.7 | 1.1 | 18.0 | 11.1 | 42.1 | 8.0 |
| $+L_{dis}^{s} + L_{dis}^{t} + L_{CE}^{t}$ | 88.1 | 46.6 | 82.1 | 30.2 | **28.4** | 39.7 | 31.3 | 38.8 | 83.6 | 30.7 | 75.1 | 61.9 | 28.5 | 84.3 | 16.3 | 36.3 | 29.1 | 25.0 | 29.4 | 46.6 | 12.5 |
| $+L_{CE}^{t} + L_{CE}^{tP}$ | 88.9 | 47.1 | 83.0 | 31.0 | 27.3 | 39.7 | 31.0 | 36.0 | 84.3 | 32.6 | 75.1 | 62.0 | 29.4 | 84.6 | 16.6 | 35.7 | 27.2 | 19.2 | 28.4 | 46.3 | 12.2 |
| CAG-UDA (Stage 1) | 88.8 | 47.5 | 83.6 | 31.7 | 29.1 | 39.7 | 34.4 | 35.6 | 84.4 | 33.0 | 76.8 | **62.1** | 28.2 | 84.5 | 17.2 | 35.2 | 32.0 | 25.8 | 27.6 | 47.2 | 13.1 |
| CAG-UDA (Stage 2) | **90.4** | 50.6 | 84.0 | 33.5 | 28.3 | **39.9** | 31.6 | 42.4 | 85.1 | 35.2 | 77.3 | 61.5 | 34.2 | 84.9 | 19.4 | 41.7 | 41.0 | 27.3 | 32.0 | 49.5 | 15.4 |
| CAG-UDA (Stage 3) | **90.4** | **51.6** | **83.8** | **34.2** | 27.8 | 38.4 | 25.3 | **48.4** | **85.4** | **38.2** | **78.1** | 58.6 | **34.6** | 84.7 | **21.9** | **42.7** | **41.1** | **29.3** | **37.2** | **50.2** | **16.1** |

**Ablation studies:** The ablation study results are listed in Table 4. We add a superscript $P$ to the symbols of losses to denote that the active target samples are identified by category probabilities as described in Section 3.3. Several models were trained by combining $L_{CE}^{t}$ with different losses. As can be seen from the $2^{nd}$ and $3^{rd}$ rows, the proposed category anchor-guided PLA is more effective than the predicted category probability-based one. More detailed comparisons of different hyper-parameters can be found in the supplement. In addition, the CE loss is more effective than the distance loss. The results in the $4^{th}$ row demonstrate the complementarity between the CE loss and distance loss, as well as between the category anchor-based and probability-based PLA. We combine them as in Eq. (12) to train the CAG-UDA model and obtain a better result as listed in the bottom row. Finally, the stagewise trained CAG-UDA model obtains an mIoU of 50.2, outperforming the SOTA models. Besides, the CAG-UDA model has been trained for an extra stage, $e.g.$, Stage 4. However, it is saturated at 50.2 mIoU with no improvement.

## 4.3 Limitations

The proposed CAG-UDA model relies on reliable pseudo-labels to guarantee a correct supervision imposed on the network to be trained. To this end, we adopt a warm-up strategy to roughly align two domains together and increase the reliability of the generated pseudo-labels by the CAs, as described in Section 3.4. In contrast, we also conducted an experiment by removing the warm-up stage and observed a significant drop of 6.3 mIoU. Some techniques can also be used to obtain reliable pseudo-labels such as enforcing local smoothness on the probability map, utilizing a normalized threshold during assigning pseudo-labels, and reducing the appearance bias through a style transfer module. We leave it as the future work to build a stage-free and end-to-end CAG-UDA model.

## 5 Conclusion

In this paper, we proposed a novel category anchor-guided (CAG) unsupervised domain adaptation (UDA) model for semantic segmentation. The CAG-UDA model successfully adapts the segmentation model to the target domain through category-wise feature alignment guided by category anchors. Specifically, we proposed a category anchor construction module, an active target sample identification module, and a pseudo-label assignment module. We utilized a distance loss and a CE loss based on the identified active target samples, which complementarily enhance the adaptation performance. We also proposed a stagewise training mechanism to reduce the error accumulation and adapt the CAG-UDA model progressively. The experiments on the GTA5 and SYNTHIA datasets demonstrate the superiority of the CAG-UDA model over representative methods on generalization to the Cityscapes dataset.

## Acknowledgements

This work is supported by the Australian Research Council Project FL-170100117 and the National Natural Science Foundation of China Project 61806062.

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
