[Supplementary Material]

# Category Anchor-Guided Unsupervised Domain Adaptation for Semantic Segmentation: Supplementary Material

**Hyper-parameter settings:** The hyper-parameter $\triangle_d$ in the proposed method has an influence on the number of the activated target samples as described in Section 3.2 in the manuscript. Hence, we investigate its influence on the segmentation accuracy. We conducted experiments at different hyper-parameter settings by leveraging the following two losses, $i.e.$, $L_{CE}^s$ and $L_{CE}^t$. The results are summarized in Table 1 and Table 2. As can be seen, the mIoU turns out to be insensitive to the setting of $\triangle_d$, $e.g.$, in the range of [2,4]. We set $\triangle_d$ to 2.5 in all the experiments if not specified.

As a reference, we also report the percentage of selected target samples by using the probability-based PLA method with a threshold $P_0 = 0.95$. It has a comparable percentage of the activated target samples with $\triangle_d = 2.5$. However, the mIoU of probability-based PLA with $P_0 = 0.95$ is lower than the proposed CAs-based PLA with $\triangle_d = 2.5$ by a large margin, $i.e.$, 1.8. It proves that the CAs-based pseudo-labels are more informative, and thus lead to better performance.

For $\lambda_1$ and $\lambda_2$, the performance usually improves when one rises and the other is fixed as shown in Table 3. The performance is also stable with respect to the changes of hyper-parameters $\lambda_1$ and $\lambda_2$ in ranges of [0.3, 1.1] and [0.3, 1.1], respectively.

More subjective results can be found in Figure 1 and Figure 2. It is obvious that CAs-based PLA assigns pseudo-labels on both small objects and large stuff, while probability-based PLA only concentrates on large stuff.

Table 1: Hyper-parameters study (GTA5→Cityscapes).

|  | hyper-paramter | value | percentage | mIoU |
|---|---|---|---|---|
| $+L_{CE}^t$ | $\triangle_d$ | 2 | 52.8% | 45.9 |
| $+L_{CE}^t$ | $\triangle_d$ | 2.5 | 40.9% | **46.1** |
| $+L_{CE}^t$ | $\triangle_d$ | 3 | 31.3% | 46.0 |
| $+L_{CE}^{tP}$ | $P_0$ | 0.95 | 39.0% | 44.3 |

Table 2: Hyper-parameters study on $\Delta_d$ (GTA5→Cityscapes).

| $\lambda_2 = 0.7, L = L_{CE}^s + \lambda_2 L_{CE}^t$ | | | | | | |
|---|---|---|---|---|---|---|
| $\Delta_d$ | 1 | 1.5 | 2 | 2.5 | 3 | 3.5 | 4 |
| mIoU | 45.19 | 45.53 | 45.93 | 46.14 | 46.01 | 45.96 | 45.67 |

| Input Image | Probability-based PLA | CAs-based PLA | Ground Truth |

Figure 1: Subjective comparison between probability-based PLA and the proposed CAs-based PLA on some images from the Cityscapes training set. Best viewed in color and zoom-in.

Table 3: Hyper-parameters study on $\lambda_1$, $\lambda_2$ (GTA5→Cityscapes).

| $\Delta_d = 2.5$, $L = L_{CE}^s + \lambda_1 ( L_{dis}^s + L_{dis}^t ) + \lambda_2 ( L_{CE}^t + L_{CE}^{tP} )$ | | | | | | | | | |
|---|---|---|---|---|---|---|---|---|---|
| $\lambda_1$ | 0.3 | 0.5 | 0.7 | 0.9 | 1.1 | 0.3 | 0.3 | 0.3 | 0.3 | 0.3 |
| $\lambda_2$ | 0.7 | 0.7 | 0.7 | 0.7 | 0.7 | 0.3 | 0.5 | 0.7 | 0.9 | 1.1 |
| mIoU | 47.13 | 47.13 | 47.18 | 47.21 | 47.19 | 47.03 | 47.16 | 47.13 | 47.23 | 47.20 |

Figure 2: Subjective comparisons between probability-based PLA and the proposed CAs-based PLA on some images from the Cityscapes training set. Best viewed in color and zoom-in.

Figure 3: Subjective comparisons between the models in different phases. All images are from the Cityscapes validation set. Best viewed in color.