[Reviews · NeurIPS 2019]

Reviewer 1



The paper presents an unsupervised domain adaptation approach for semantic segmentation that uses category centroids (anchors) for category-wise feature alignment, trying to keep features from the same category nearby while keeping features from other categories far apart. Pseudo-labeling is used on target samples when sufficiently close to a source anchor (active samples). Pros: + To my knowledge, this is the first UDA approach for semantic segmentation that combines pseudo-labeling with feature alignment. Works applying similar ideas exist for classification [3,17,37], but the differences are sufficient and well acknowledged by the authors. + The presented model is sound and well described. Fig. 1 is helpful to understand the intuition behind the idea (b-c) as well as the actual architecture (a). + Sensible stage-wise training procedure to guarantee good initial anchors, although it makes the training more cumbersome. Also, it seems that it is not saturated in stage 3, would the results improve if trained for more stages? + Convincing results, especially for small classes in which CA-based PLA clearly seems to be crucial in providing good pseudo-labels. Moreover, the authors provide a possible explanation of the limitation of their method with respect to style-transfer for stuff classes. As a suggestion for possible future work, one could think in combining CAG with a style-transfer module to address stuff classes as well. Cons: - Since pseudo-labeling is done at the pixel level, the pseudo-labels are not necessarily very smooth. The shown pseudo-labels seem relatively smooth (fig. 2 and suppl.), but the method could benefit from enforcing local smoothness to increase the robustness of ATI or PLA. - It would be clearer to add the explicit definition of the loss in L208. - Missing related work (although for classification): "Unsupervised Domain Adaptation with Similarity Learning", Pinheiro, CVPR2018. Minor things - Citation missing in L98 for Li et al. - In eq. 4, and if x is defined as the image like in L128, the input of f_D should be something like Enc(x) (i.e. the encoded features) for coherence with Fig.1a. I understand this is a notation abuse for clarity, but this should be mentioned somewhere - A few typos: SYNTHIA in Tab.2's caption, outperforms in L253, etc. The authors addressed most of the reviewers's concerns in the rebuttal and thus I keep my acceptance score.

Reviewer 2



There are some interesting novel concepts introduced in the UDA framework proposed by the authors. Overall the paper is clearly written. I feel the paper could be improved by demonstrating the effect of using active samples, as these seem to be a prominent elements in the optimisation framework. The ablation results in Table 3 also indicate that +L_CE^tP and L_CE^t are both significant contributors to the final performance. I'd like to see more elaboration on the setting of Delta_d (used both in Eq.5 and 7), which surely decides the number of active samples in Source and Target domains - how some pseudo-label alignment may contribute/impact the final performance? What is the effect of changing the weights for Eq.11. Regarding the setting of Delta_d, which is set on the distance differences. Is this ideal? Depending on different datasets, the distance-based threshold can be quite changeable and hard to configure. Would it be more reasonable to assign a threshold in a normalised setting? Also the same threshold is used in both Source and Target anchors, which may not be optimal? Some minor corrections: The context around Eq.5 can be written as - Mathematically, this can be formulated as follows. We first define the distance between ... and the c^th category anchor as d_ijc^t =... (5) Then, we sort .... in an ascending order, and compare the shortest d_ijc with the second shortest d_ijc... .... we identify this target sample as [an] active one, ... It is better to also give the formula for L_CE^tP, given its importance. The comments right under (7) seem a bit loose. You said "they turn out to be more reliable than ...", without providing any evidence or justification. What do you mean by "they do not depend on the decision boundaries"?

Reviewer 3



In this paper, a category anchor based domain adaptation approach is proposed for semantic segmentation. The centroid of each source domain category is used as anchor for discovering confident target samples, in which difference of distances to anchors is used as the metric. Identified target samples are then labeled according to its closest anchor. Then the pseudo-labeled target samples are used for training the segmentation model. Distances to anchors are also used as a regularization to guide the training. Experiments on benchmark datasets validate the effectiveness of the proposed method. The paper is well written and easy to follow. One problem with the proposed method is that it involves many hyperparameters, for example, $delta_d$, $\lambda_1$ and $\lambda_2$. Although ablation study is provided in supplementary, it is still quite limited. It would be more convincing if a wider range of vlidatation on the hyperparameters are provided. How important is the warm-up stage? What if you remove it?

[Author Response · NeurIPS 2019]

Table 1: Hyper-parameters study on $\Delta_d$ ( GTA5→Cityscapes).

| $\lambda_2 = 0.7$, $L = L_{CE}^s + \lambda_2 L_{CE}^t$ | | | | | | |
|---|---|---|---|---|---|---|
| $\Delta_d$ | 1 | 1.5 | 2 | 2.5 | 3 | 3.5 | 4 |
| mIoU | 45.19 | 45.53 | 45.93 | 46.14 | 46.01 | 45.96 | 45.67 |

Table 2: Hyper-parameters study on $\lambda_1$, $\lambda_2$ ( GTA5→Cityscapes).

| $\Delta_d = 2.5$, $L = L_{CE}^s + \lambda_1 ( L_{dis}^s + L_{dis}^t ) + \lambda_2 ( L_{CE}^t + L_{CE}^{tP} )$, stage 1 | | | | | | | | | |
|---|---|---|---|---|---|---|---|---|---|
| $\lambda_1$ | 0.3 | 0.5 | 0.7 | 0.9 | 1.1 | 0.3 | 0.3 | 0.3 | 0.3 | 0.3 |
| $\lambda_2$ | 0.7 | 0.7 | 0.7 | 0.7 | 0.7 | 0.3 | 0.5 | 0.7 | 0.9 | 1.1 |
| mIoU | 47.13 | 47.13 | 47.18 | 47.21 | 47.19 | 47.03 | 47.16 | 47.13 | 47.23 | 47.20 |

We appreciate the valuable comments from all reviewers and will carefully take them into account in the final version.
In this rebuttal, we focus on major concerns. The source code will be released for verification and reproducibility.

**R1.1** Stage-wise training procedure makes training cumbersome. Will the results improve if trained for more stages?

⇒ We tried to train the CAG model in a single stage and update the pseudo-labels at each iteration. However, it is not
stable because there are some error-prone pseudo-labels, which may produce incorrect supervision signals, lead to more
erroneous pseudo-labels iteratively, and trap the network to a local minimum with poor performance eventually, *i.e.*,
less than 30 mIoU. To address this issue, we used the stage-wise training procedure by fixing the anchors at each stage.
It is noteworthy that we reduced the training epochs at each stage, *e.g.*, 20 epochs, so that the overall training cost is
comparable to previous methods. Nevertheless, we are open to explore more efficient pseudo-labels assignment and
training techniques to further accelerate the training procedure. Besides, after the CAG model is trained for an extra
stage, it reaches 50.21 mIoU, which is saturated compared with stage 3.

**R1.2** The shown pseudo-labels seem relatively smooth (fig. 2). Enforcing local smoothness can benefit the method.

⇒ The pixels in images are spatially coherent, and the features $f_D$ from the penultimate layer are expected to have
effective characteristics for clustering. Therefore, the pseudo-labels by the proposed CAs-based assignment are
smooth. Enforcing local smoothness can be promising for enhancing the robustness of ATI or PLA and the accuracy of
pseudo-labels, *e.g.*, applying CRFs on the CAs-based distance map.

**R1.3**, **R2.4** It would be clearer to add the explicit definition of the loss $L_{CE}^{tP}$.

⇒ The explicit definition of $L_{CE}^{tP}$ is given by $L_{CE}^{tP} = - \sum_{i=1}^{M} \sum_{j=1}^{H \times W} a_{ij}^{tP} \sum_{c=1}^{C} \hat{y}_{ijc}^{tP} log \left( p_{ijc}^{tP} \right)$ similar to $L_{CE}^t$ in
Eq.(9), where $a_{ij}^{tP}$, $y_{ijc}^{tP}$ and $p_{ijc}^{tP}$ refer to the active state, assigned pseudo-labels vectors and network output respectively,
which will be added in the final version.

**R2.1**, **R3.1** It would be more convincing if a wider range of validation on the hyper-parameters is provided.

⇒ We investigated the effect of a wider range of hyper-parameters including $\Delta_d$, $\lambda_1$ and $\lambda_2$. The results are listed in
Tables 1 and 2. We can see that the performance peaks at $\Delta = 2.5$, and is not sensitive to the choice of $\Delta$ in the range of
[2, 3.5]. For $\lambda_1$ and $\lambda_2$, the performance usually improves when one rises and the other is fixed. The performance is
also stable with respect to the changes of hyper-parameters $\lambda_1$ and $\lambda_2$ in ranges of [0.3, 1.1] and [0.3, 1.1], respectively.

**R2.2** Is it more reasonable to assign $\Delta_d$ in a normalized setting? The same $\Delta_d$ in both domains may not be optimal?

⇒ We used the distance-based threshold in the proposed CAs-based PLA because it is simple and easy to implement.
It is noteworthy that we used the same threshold for all categories to reduce the number of free hyper-parameters. It
is empirically effective and usually performs well. We did not use the threshold in the source domain, because the
ground-truth labels are available. In the final version, we will take this suggestion by comparing the current setting with
the normalized threshold.

**R2.3** No evidence on "they turn out to be more reliable"; clarification of "they do not depend on the decision boundaries".

⇒ Due to the lack of the target domain labels, the classifier is biased to the source domain and does not generalize
well to the target domain, as shown in Fig.1 (c). Consequently, some of the pseudo-labels from predicted probabilities
may be erroneous. Based on the observation of the intra-category clustering characteristics, we propose the CAs-based
assignment method which is independent of the classifier and the biased probabilities. Under the same setting of the
experiments, using the pseudo-labels assigned by anchors achieves better performance than those assigned by predicted
probabilities by a large margin. With this regard, we claim that "CAs-based pseudo-labels are more reliable and do not
depend on the decision boundary". Here the decision boundary refers to the classification hyperplane formed by the
output layer of the network.

**R3.2** How important is the warm-up stage? What if you remove it?

⇒ The distance loss and CE loss used in Eq.(11) rely on reliable pseudo-labels to guarantee a correct supervision
imposed on the network. Adding the warm-up stage can roughly align both domains (feature distributions) and increase
the reliability of the pseudo-labels by the CAs-based PLA. In our experiments, we find that removing the warm-up
stage leads to a significant performance drop, *e.g.*, 6.3 mIoU. Nevertheless, we agree that it is valuable (1) to explore
more effective PLA and training techniques to further improve the reliability of pseudo-labels and (2) to reduce the side
influence from the error-prone ones, *e.g.*, using a progressive PLA based on adaptive thresholds and combining CAG
with a style-transfer module suggested by reviewers #1 and #2.

[Meta-Review · NeurIPS 2019]

The paper presents a novel approach for unsupervised domain adaptation, which employs category-wise feature alignment and self-supervised training with pseudo-features. An active target sample selection strategy is proposed leveraging distance category anchors for pseudo labeling. Overall the approach is clearly presented, convincing, and well supported by empirical evaluation. The reviewers and AC have examined the authors feedback, which satisfactorily addresses the points raised in the reviews. We strongly recommend that the authors incorporate this feedback in the revised paper.